



# A NEMO-based model of *Sargassum* distribution in the Tropical Atlantic: description of the model and sensitivity analysis (NEMO-Sarg1.0)

Julien Jouanno[1], Rachid Benshila[1], Léo Berline[2], Antonin Soulié[1], Marie-Hélène Radenac[1], Guillaume Morvan[1], Frédéric Diaz[2], Julio Sheinbaum[3], Cristele Chevalier[2], Thierry Thibaut[2], Thomas Changeux[2], Frédéric Menard[2], Sarah Berthet[4], Olivier Aumont[5], Christian Ethé[5], Pierre Nabat[4], Marc Mallet[4].

[1] LEGOS, Université de Toulouse, IRD, CNRS, CNES, UPS, Toulouse, France
[2] Aix-Marseille University, Université de Toulon, CNRS/INSU, IRD, MIO UM 110, Mediterranean Institute of Oceanography (MIO), Campus of Luminy, 13288 Marseille, France
[3] CICESE, Ensenada, Mexico.
[4] CNRM, Université de Toulouse, Météo-France, CNRS, Toulouse, France
[5] Laboratoire d'Océanographie et de Climatologie: Expérimentation et Approches Numériques, IRD-IPSL, 4 Place Jussieu, 75005 Paris, France

*Correspondence to*: Julien Jouanno (julien.jouanno@ird.fr)

**Abstract.** The Tropical Atlantic is facing a massive proliferation of Sargassum since 2011, with severe environmental and socioeconomic impacts. The development of Sargassum modelling is essential to clarify the link between Sargassum distribution and environmental conditions, and to lay the groundwork for a seasonal forecast on the scale of the Tropical Atlantic basin. We developed a modelling framework based on the NEMO ocean model, which integrates transport by currents and waves, physiology of Sargassum with varying internal nutrients quota, and considers stranding at the coast. The model is initialized from basin scale satellite observations and performance was assessed over the Sargassum year 2017. Model parameters are calibrated through the analysis of a large ensemble of simulations, and the sensitivity to forcing fields like riverine nutrients inputs, atmospheric deposition, and waves is discussed. Overall, results demonstrate the ability of the model to reproduce and forecast the seasonal cycle and large-scale distribution of Sargassum biomass.

## 1 Introduction

The massive development of holopelagic *Sargassum* spp. in the Northern Tropical Atlantic Ocean from 2011 causes annual stranding in millions of tons on the coasts of the Lesser Antilles, Central America, Brazil and West Africa (e.g., Smetacek and Zingone 2013, Wang and Hu 2016, Langin 2018, Wang et al. 2019). The proliferation affects the whole northern Tropical Atlantic area as illustrated by satellite observations for summer 2017 (Figure 1, Berline et al. 2020).



Modelling and forecasting the *Sargassum* proliferation and strandings is essential for designing effective integrated risk management strategies, and is a strong and pressing demand from the civil society. This operational challenge concerns both event forecasts (i.e., on a one-week scale) and long-term forecasts (one to several months). While many efforts have been made for short-term forecast, initiatives for reliable long-term forecasting are very scarce and face several scientific challenges such

as the large uncertainties in *Sargassum* detection and biomass quantification (Wang et al. 2018, Ody et al. 2019), a lack of knowledge on *Sargassum* physiology, and last but not least the absence of tools specifically designed to reproduce the large-scale distribution and evolution of these macro-algae.

Recent studies suggest that the increasing incidence of *Sargassum* blooms and its year-to-year variability is multifactorial: it may result from riverine and atmospheric fertilization of the upper ocean, West Africa upwelling variability, vertical

exchanges at the mixed-layer base in the region of the Inter-Tropical Convergence Zone, ITCZ) or anomalous transport due to climate variability (Oviatt et al. 2018, Wang et al. 2019, Johns et al. 2020). This highlights the complexity of the phenomenon and the need for a basin-scale and inter-disciplinary approach.

In the recent years, modelling effort mainly focused on the transport properties of *Sargassum* rafts by offshore currents (Wang and Hu, 2017, Brooks et al. 2018, Maréchal et al., 2017, Putman et al 2018, 2020, Wang et al. 2019, Berline et al. 2020,

Beron-Vera and Miron, 2020), with significant advances on the role of inertia in the drift trajectories (Brooks et al. 2019, Beron-Vera and Miron 2020) and the importance of considering windage to properly resolve the drift of the *Sargassum* mats (Putman et al. 2020, Berline et al. 2020). To our knowledge, Brooks et al. (2018) were the first to integrate *Sargassum* physiology along the trajectories and showed that considering growth and mortality improved the modelling of the large-scale distribution of *Sargassum*. A similar result was obtained in Wang et al. (2019), although they did not consider directly the

physiology of the algae but local growth rate based on satellite observations. Indeed, few studies have investigated the biology and ecology of this holopelagic *Sargassum* species that proliferate in the Atlantic and their response to the variability of environmental parameters (Lapointe 1995, 1986, Hanisak and Samuel 1987, Carpenter and Cox 1974, Hanson 1977, Howard and Menzies 1969).

In the present paper, we describe the numerical model we developed to represent the distribution of holopelagic *Sargassum*.

This model relies on an Eulerian approach and integrates both transport and a simplified physiology model of the macroalgae. It is based on the NEMO modelling system, which is widely used by the research community and European ocean forecasting centers (*e.g.*, Mercator Ocean International, ECMWF), allowing efficient parallelization and interfacing with physical-biogeochemical models. In the following section, we review current knowledge on the ecology of *Sargassum*. The modelling system is described in section 3. Section 4 shows the performance of the model at seasonal scale and discusses sensitivity of

the modeled *Sargassum* distribution to the forcing fields. Discussion and a summary are given in the final Section.





## 2 Physiological and ecological features of holopelagic *Sargassum*

Pelagic *Sargassum* species (to date *Sargassum natans* and *S. fluitans*) are brown algae (Phaeophyceae) that live at the surface of the ocean, never attached to any substrate. Within these two taxonomic groups, three types of *Sargassum* that can be distinguished according to morphological features, appear to fuel the recent *Sargassum* inundations in the Caribbean: *S. fluitans* III, *S. natans* I, *S. natans* VIII (Schell et al 2015). We still lack knowledge on the distribution of these species, but in recent years, *S. fluita*ns III was predominant in 2017 (Ody et al., 2019) and form beaching on the Yucatan Coast, comprising on average >60 % of total wet biomass (Garcia Sanchez et al. 2020), whereas Schell et al. (2015) reported a predominance of *S. natans* VIII in 2015.

One individual *Sargassum* fragment can vary in length from just one cm to more than 1 m. Under the action of Langmuir cells and ocean currents, *Sargassum* tends to group together to form large floating rafts at the water surface (e.g., Langmuir 1936, Zhong et al. 2012). Individuals in these aggregations can be easily dispersed by wind, waves or any event (Ody et al., 2019). These assemblages spread out horizontally and can reach several tens of km and a few meters thick.

Biological and physiological features are species dependent. We know relatively little about the physiology of these *Sargassum*. Considering biomass, their maximum growth rate is estimated to be around 0.1 d$^{-1}$ (Lapointe 1986, Hanisak and Samuel 1987, Lapointe et al. 2014). The *Sargassum* growth is sensitive to light and temperature. Carpenter and Cox (1974) suggest light saturation under normal October light conditions in the Sargasso Sea (35 W m$^{-2}$), while Hanisak and Samuel (1987) found a higher saturation range of ~43-65 W m$^{-2}$. The temperature dependence in Hanisak and Samuel (1987) for *Sargassum natans* suggests a broad optimal temperature range of 18-30°C and indicates no growth at 12°C. We lack information on *Sargassum fluitans* response to the variability of the environmental parameters.

Lapointe (1986) highlights a growth mainly limited by phosphate availability, while the presence of nitrifying epiphytes (Carpenter 1972, Michotey et al. 2020) could be a non-negligible source of nitrogen for *Sargassum*, as could urea and ammonium excreted by fish (Lapointe et al. 2014). It is also likely that *Sargassum* are able to store some nutrients in their tissues, as do other brown algae (e.g., Hanisak 1983). This hypothesis is supported by Lapointe et al. (1995) whose measurements revealed variable elemental compositions between individuals sampled in neritic *vs* oceanic waters. In addition, no macro-herbivores control holopelagic development offshore by grazing (Butler et al. 1983).

## 3 The *Sargassum* modelling framework

Our modelling strategy relies on a physical-biogeochemical model that resolves currents and nutrient variability in the Atlantic. We choose to develop a regional configuration so the model can be tuned to the region specificity and can be used to perform sensitivity tests as discussed in Section 4. In addition, our approach is based on a *Sargassum* model that integrates transport, stranding and physiology of the macroalgae in the ocean surface layer, forced with surface fields obtained from the physical-biogeochemical model.





The two models share the same horizontal domain and grid, and are both based on the NEMO modelling system version 4.0 (Nucleus for European Modelling of the Ocean, Madec et al. 2016). They are not coupled assuming then that *Sargassum* does not compete with phytoplankton (and heterotrophic bacteria) for nutrient resources, and that they are not grazed by the

herbivore compartments of the biogeochemical model.

### 3.1 The physical-biogeochemical model TATL025BIO

For the physical component of the simulation, we use the regional NEMO-based configuration described in Hernandez et al. (2016, 2017) and Radenac et al. (2020) that covers the tropical Atlantic between 35°S and 35°N and from 100°W to 15°E. The resolution of the horizontal grid is ¼° and there are 75 vertical levels, 24 of which are in the upper 100 m of the ocean. The

depth interval ranges from 1 m at the surface to about 10 m at 100 m depth. Interannual atmospheric fluxes of momentum, heat, and freshwater are derived from the DFS5.2 product (Dussin et al., 2016) using bulk formulae from Large and Yeager (2009). Temperature, salinity, currents, and sea level from the MERCATOR global reanalysis GLORYS2V4 (Storto et al., 2018) are used to force the model at the lateral boundaries.

The physical model is coupled to the PISCES (Pelagic Interaction Scheme for Carbon and Ecosystem Studies)

biogeochemical model (Aumont et al., 2015) that simulates the biological production and the biogeochemical cycles of carbon, nitrogen, phosphorus, silica, and iron. We use the PISCES-Q version with variable stoichiometry described in Kwiatkowski et al. (2018), and with an explicit representation of three phytoplankton size classes (picophytoplankton, nanophytoplankton and microphytoplankton) and two zooplankton compartments (nanozooplankton and mesozooplankton). The model also includes three non-living compartments (dissolved organic matter and small and large sinking particles). The biogeochemical model is

initialized and forced at the lateral boundaries with dissolved inorganic carbon, dissolved organic carbon, alkalinity, and iron obtained from stabilized climatological 3-D fields of the global standard configuration ORCA2 (Aumont and Bopp, 2006), and nitrate, phosphate, silicate, and dissolved oxygen from the World Ocean Atlas observation database (WOA; Garcia et al., 2010). The model is run from 2006 to 2017 and daily fields are extracted to force the *Sargassum* model.

Particular care has been given to the prescription of the atmospheric and riverine fluxes of nutrients. The river runoffs

are based on daily fluxes from the ISBA-CTRIP reanalysis (Decharme et al. 2019) which has proven to accurately reproduce the interannual variability of the large rivers of the basin (e.g., see Giffard et al. 2019 for the Amazon river). The riverine nutrient fluxes concentrations are from the GLOBAL-NEWS2 dataset, corrected with *in situ* observations from the Amazon basin water resources observatory database (HYBAM, https://hybam.obs-mip.fr/) for the Amazon, Orinoco and Congo rivers. As in Aumont et al. (2015), we consider an atmospheric supply of P, Fe and Si from dust deposition. Here, these fluxes are

forced using monthly dry-plus-wet deposition products (DUDPWTSUM) from the MERRA-2 data available on the NASA Giovanni website (http://disc.sci.gsfc.nasa.gov/giovanni). Comparison with *in situ* observations of dust fluxes in Guyana and in Barbados lead to an excellent match with MERRA2 fluxes (Prospero et al. 2020). A climatological deposit of N is obtained from global climate simulations carried out with the ARPEGE-Climate model (Michou et al. 2020). An interactive aerosol





scheme including nitrate and ammonium particles (Drugé et al. 2019) is included in ARPEGE-Climate, allowing us to produce fields of wet and dry deposition of nitrogen, ammonium and ammonia. Figure 2 illustrates that dust and nitrogen fluxes to the ocean are strong in our region of interest, and most particularly in the ITCZ region where atmospheric convergence may focus the wet fluxes.

The modeled chlorophyll for year 2017 is compared with GLOBCOLOUR satellite estimates of chlorophyll for the same year (Figure 3a,b). Model $NO_3$ and $PO_4$ concentrations for 2017 are compared with historical *in situ* measurements (Figure 3c-f) from the GLODAPV2 database (Olsen et al., 2016). The model reproduces the major chlorophyll structures and in particular the contrast between the oligotrophic subtropical gyre and productive coastal and equatorial upwellings (Figures 3a, b). As many other models (*e.g.,* see the CMIP6 model evaluation by Seferian et al. 2020), it struggles to reproduce the offshore extent of the large river plumes and Guinea dome productivity. Moreover, coastal upwellings tend to be too productive offshore or downstream (for the equatorial upwelling). But above all, it represents realistically the chlorophyll distribution in the region of the ITCZ (~0-10°N) and in the Caribbean Sea. As observed, nitrate concentrations are high in the upwelling areas (Figures 3c, d) but weaker than observed off these regions. It is worth noticing that historical observations of surface nitrate concentrations in the tropical band show very heterogeneous and contrasted values between cruises, so the reliability of a nitrate climatology in this area remains uncertain (Figure 3c). The model reproduces realistically the observed interhemispheric gradient of surface phosphate concentrations even though it likely overestimates areas of high phosphate contents (Figures 3e, f).

## 3.2 The *Sargassum* model NEMO-Sarg1.0

The *Sargassum* model relies on the strategy used to represent the distribution of other macroalgae species (*e.g.*, Martin and Marques 1993, Solidoro et al. 1997, Perrot et al. 2014 or Ren et al. 2014). At the difference of these previous works, we also consider transport by 2D advection/diffusion equations and sink due to stranding at the coast. Growth is modeled as a function of internal reserves of nutrients, dissolved inorganic nutrients in the external medium, irradiance and sea temperature. As the eco-phycological features are species dependent, the actual knowledge on the three morphotypes of holopelagic *Sargassum* does allow to discriminate between them. Following the formalism given in Ren et al. (2014), the physiological behavior is described from three state variables: the contents in carbon (C), nitrogen (N) and phosphorus (P), with local variations reflecting the difference between uptake and loss rates.

$$\frac{\partial C}{\partial t} = U_C - \phi_C$$

$$\frac{\partial N}{\partial t} = U_N - \phi_N$$

$$\frac{\partial P}{\partial t} = U_P - \phi_P$$


where $U_C$, $U_N$ and $U_P$ are the uptake rates of carbon, nitrogen and phosphorus respectively, and $\Phi_C$, $\Phi_N$, $\Phi_P$ the loss rates.

The rate of carbon uptake reads as follows: $U_c = C \cdot \mu_{max} \cdot f[T] \cdot f[I] \cdot f[Qn] \cdot f[Qp]$, with $\mu_{max}$ the maximum net carbon growth rate, and the four subsequent terms standing for uptake limitation by temperature ($T$), solar radiation ($I$), N-quota ($Qn$), and P-quota ($Qp$), respectively. N and P quotas represent the ratios of nitrogen and phosphorus to carbon in the organism and are computed as N/C and P/C respectively. The C content (C) can be directly converted to dry biomass

considering a mean carbon to dry weight ratio of 27% (Wang et al. 2018).

The temperature dependence is adapted from Martins and Marques (2002):

$$f(T) = e^{-\frac{1}{2}\left(\frac{T-Topt}{T_x-T}\right)^2}$$


with $\{T_x = T_{min} \; for \; T \leq T_{opt} \; ; \; T_x = T_{max} \; for \; T > T_{opt}\}$. $T_{opt}$ is the optimum temperature at which growth rate is maximum, $T_{min}$ is the lower temperature limit below which growth ceases, $T_{max}$ is the upper temperature limit above which growth ceases. Such function aims at representing a broad optimal temperature range as suggested by experiments in Hanisak and Samuel (1987). The dependence to light follows the function given in Martins and Marquez (2002) with photoinhibition

at high light:

$$f(I) = \frac{I}{I_{opt}} \cdot e^{\left(1-\frac{I}{I_{opt}}\right)}$$

We have very few information on the response curve relating the nutrient quota to *Sargassum* growth but experiments for brown seaweeds suggest hyperbolic relationship (*e.g.*, Hanisak 1983). So, the dependence to the internal nitrogen and

phosphorus pools are computed as a hyperbolic curve controlled by the minimum and maximum cell quota:

$$f(Q_N) = \left(\frac{1-Q_{Nmin}/Q_N}{1-Q_{Nmin}/Q_{Nmax}}\right)$$
$$f(Q_P) = \left(\frac{1-Q_{Pmin}/Q_P}{1-Q_{Pmin}/Q_{Pmax}}\right)$$

The nitrogen and phosphorus uptake rates depend on the nitrogen ($V_{Nmax}$) and phosphorus ($V_{Pmax}$) maximum uptake velocities, a Monod kinetic that relates uptake to nutrient concentrations in the water, and a function of quota which aims at representing downregulation of the transport system for N and P when approaching the maximum quotas (Lehman et al. 1975):





$$U_N = V_{Nmax} \cdot C \cdot \left(\frac{[N]}{K_N + [N]}\right) \cdot (\frac{Q_{Nmax} - Q_N}{Q_{Nmax} - Q_{Nmin}})$$

$$U_P = V_{Pmax} \cdot C \cdot \left(\frac{[P]}{K_P + [P]}\right) \cdot (\frac{Q_{Pmax} - Q_P}{Q_{Pmax} - Q_{Pmin}})$$

The carbon loss aims at representing mortality, stranding and sinking:

$$\phi_C = C \cdot \left(\frac{m}{e^{-\lambda_m \cdot (T-30°C)}} + \delta_{land} \cdot \alpha_{grnd} + \frac{m_{LC}}{e^{-\lambda_{mLC} \cdot (H_{LC}-100m)}}\right).$$

The mortality term depends on $m$ the mortality rate, $\lambda_m$ a mortality coefficient, and temperature T. We thus represent thallus senescence and bacterial activity as a growing function of temperature (Bendoricchio et al. 1994, Ren et al., 2014).

The stranding is function of $\alpha_{grnd}$ which is a rate of *Sargassum* stranding per unit of time, and $\delta_{land}$ which is defined as follows:

$$\begin{cases} \delta_{land} = 1 \text{ if model grid cell is adjacent to two or more pixels of land,} \\ \delta_{land} = 0 \text{ otherwise.} \end{cases}$$

The sinking rate of *Sargassum* is estimated as a function of the Langmuir cell length scale $H_{LC}$ and a sinking coefficient $m_{LC}$. The aim is to reproduce possible *Sargassum* loss by Langmuir cell as hypothesized by Johnson and Richardson (1977) or Woodcock (1993) to explain large amounts of *Sargassum* observed at the sea floor (Schoener and Rowe 1970, Baker et al. 2018). Following, Axel et al. (2014), we estimate $H_{LC}$ as the depth that a water parcel with kinetic energy $u_s^2/2$ can reach on its own by converting its kinetic energy to potential energy. This corresponds to: $-\int_{-H_{LC}}^{0} N^2(z) \cdot z \cdot dz = \frac{1}{2} \cdot u_s^2$, with $u_s$ the Stokes drift and $N^2$ the Brunt Vaisala frequency. We fixed a 100m depth threshold as *Sargassum* becomes massively negatively buoyant at these depths (Johnson and Richardson, 1977).

Losses of nitrate and phosphate are function of the loss of biomass and internal N and P quotas:

$$\phi_N = \phi_C \cdot Q_N , \phi_P = \phi_C \cdot Q_P .$$

The transport of C, N, and P is resolved using 2D advection/diffusion equations discretized on a grid at 1/4° resolution with a single vertical layer representing a surface layer of water of one-meter depth. The surface velocities used for the transport, account for surface currents, windage effect and wave transport by stokes drift:





$$\phi_{transport}(Nutrient) = -U \cdot \frac{\partial Nutrient}{\partial x} - V \cdot \frac{\partial Nutrient}{\partial y} + K_h \cdot \nabla_h^2 Nutrient, \text{ with}$$


$$U, V = (u_{NEMO}, v_{NEMO}) + \alpha_{win} \cdot (u_{10m}, v_{10m}) + (u_{Stokes}, v_{Stokes})$$

where ($u_{NEMO}$,$v_{NEMO}$) are the horizontal velocity obtained from the physical-biogeochemical model, $\alpha_{win}$ is a windage coefficient, ($u_{10m}$, $v_{10m}$) the components of the wind field at 10m above the sea level, ($u_{stokes}$,$v_{stokes}$) the stokes velocity, and $K_h$ a diffusion coefficient.

## 3.3 Optimization and sensitivity experiments


The model simulations are performed for year 2017 because basin scale *Sargassum* fractional coverage observations from MODIS were available (Berline et al. 2020), with concurrent observations carried out during two cruises in the Tropical Atlantic (Ody et al. 2019).

### 3.3.1 Initialization

The simulations are initialized using January *Sargassum* mean fractional coverage, converted into dry weight biomass considering a surface density of 3.34 kg/m² and then into carbon content C considering a mean carbon to dry weight ratio of 27% (Wang et al. 2018). The initial N and P contents in *Sargassum* are derived from the initial C content and N- and P- quotas computed as the averaged values between their respective minimum values ($Q_{Nmin}$, $Q_{Nmin}$) and maximum values ($Q_{Nmax}$, $Q_{Nmax}$). The transport is forced by daily velocities from TATL025BIO simulations (see Section 3.1), 3-hours winds from the DFS5.2

dataset (Dussin et al. 2016) and stokes velocities from the ERA5 reanalysis. Daily temperature, available irradiation, and Langmuir depth were also obtained from TATL025BIO. The seawater concentrations in [N] and [P] were obtained from TATL025BIO as the sum of $NO_3$ and $NH_4$ for [N], and $PO_4$ for [P], in the top surface layer.

### 3.3.2 Ensemble strategy

The *Sargassum* model is controlled by a large number (n=18) of physiological and physical parameters for which large

uncertainties exist or most often have not been measured for the *Sargassum* species considered here. An ensemble approach has been adopted to adjust the set of parameters. We produced 10 000 sets of parameters with uniform distribution obtained from latin hypercube sampling with multi-dimensional uniformity (Deutsch and Deutsch, 2012). These sets of parameters are generated on ranges of values obtained from the literature, when available (Table 1).

The range of maximum growth rate is derived from Lapointe et al. (2014) who observed maximum growth rates of

*Sargassum fluitans* and *Sargassum natans* in neritic waters between 0.03 and 0.09 doubling d[-1]. The set of parameters for temperature limitation is mainly derived from the results of Hanisak and Samuel (1987). In particular, the $T_{max}$ range (40-50°C) is chosen to have a slight decrease of the limitation term at $T>T_{opt}$ as observed in Hanisak and Samuel (1987). The N- and P-quota are based on the observations by Lapointe et al. (1995) from which we can estimate that in average C/N ratio vary





between 20 and 30 in neritic waters and between 40 and 70 in oceanic waters, while C/P ratios vary between 200 and 500 in
neritic waters and 700 and 1000 in oceanic waters. The lower and upper limit for the maximum nitrate uptake rate ([5.0 $10^{-4}$,
1.3  $10^{-3}$] $mg\ N\ (mg\ C)^{-1}\ d^{-1}$) is estimated from measurements by Lapointe et al. (1995). From this study we estimate
maximum carbon uptake rates in neritic water at ~2 mg C (g dry wt)$^{-1}$ h$^{-1}$ with C/N ratio of 20, and maximum carbon uptake
rates in oceanic water at ~1 mg C (g dry wt)$^{-1}$ h$^{-1}$ with C/N ratio of 60. Since the model does not take into account a diurnal
cycle of light, the maximum uptake has been divided by 3 in order to have a daily-mean maximum uptake. This ratio of 3 is
obtained by comparing instantaneous gross production at full irradiance vs. measured gross production from culturing the
Sargassum under natural irradiance in Lapointe et al. (2014). Similarly, the lower and upper limits for the maximum
phosphorus uptake rate ([9 $10^{-5}$,7 $10^{-4}$] $mg\ N\ (mg\ C)^{-1}\ d^{-1}$) are estimated from measurements by Lapointe et al. (1995) which
give a C/P ratio of 200 in neritic waters and 800 in oceanic waters.

### 3.3.3 Likelihood function

The likelihood function (L) to be minimized is the Centered Root Mean Square error between monthly series of observed and
modeled *Sargassum* biomass contained in the North Tropical Atlantic Ocean, defined as [98°W-10°W, 0°-30°N], hereinafter
NTAO:

$$L = \left[\frac{1}{Nt}\sum_{t=1}^{Nt}[(\psi_m(t) - \overline{\psi_m}) - (\psi_{obs}(t) - \overline{\psi_{obs}})]^2\right]^{1/2},$$


with Nt the number of outputs (12 months in our case), $\psi_m(t)$ and $\psi_{obs}(t)$ the modeled and observed biomass in the NTAO,
and the overbars designating the annual average.

The choice of this likelihood function exerts no constraints on the spatial distribution of the *Sargassum* and does not
consider uncertainties in the satellite measurements due to false detection, cloud masking, or *Sargassum* immersion. But as
shown in the following section, such a simple strategy allows to efficiently select a set of parameters that allow a good
representation of the seasonal *Sargassum* distribution.

### 3.3.4 Sensitivity analysis

Once a set of values minimizing *L* was found, one-at-a-time sensitivity experiments were also performed, where only one
single parameter is varied by ±10% in two model runs while the others are fixed (e.g., Ren et al. 2014, Perrot et al. 2014). This
allows to capture the direct contribution from each parameter to the output variance, with parameters varying within an
acceptable range. The set of values for the fixed parameters are given in Table 1 and were taken from the simulation with
higher *L*. Following Ren et al. (2014), deviation from the baseline simulation (D) is quantified as:





$$D = \frac{1}{N} \sum_{t=1}^{t=12} \frac{\psi_{m(i,t)} - \psi_{m(t)}^0}{\psi_{m(t)}^0}, \text{ with } N=12$$


where $\psi_{mt}^0$ refers to the simulated biomass in the NTAO from the baseline parameter set at month t, and $\psi_{m(i,t)}$ is the simulated biomass with one perturbed parameter i at month t. Two model runs were conducted with plus and minus 10% change to the baseline value.

## 4. Results

### 4.1 Seasonal distribution and sensitivity to model parameters

The observed and model seasonal distributions of *Sargassum* in 2017 are shown in Figure 4. Model distribution is obtained from the ensemble mean of the 100 simulations with the lowest *L*. Hereafter, this ensemble will be referred to as $\Omega_{100}$. Averages over selected areas are shown in Figure 5. At this stage, it is worth recording that the selection of the ensemble simulations is performed without constraints on the spatial distribution, the only constraint being on the basin scale seasonal biomass average

(*L*).

Initialized in January, the model reproduces the seasonal distributions of *Sargassum* fairly closely. It simulates the *Sargassum* drift toward the Caribbean Sea, with a summer peak of biomass followed by a decrease until the end of the year, although the increase of biomass comes a little too early in February-March. In the Caribbean Sea, the largest abundance of *Sargassum* in the Northern part of the basin near the Greater Antilles compared to the South of the Caribbean is consistent

with observations. Despite a bloom that is occurring too early (May-June) near the Lesser Antilles, the modelled seasonal cycle is also consistent with the observations in this area (Figure 5b). This is encouraging from the perspective of predicting strandings on the Caribbean islands. It also succeeds in maintaining the biomass in the Eastern part of the North Tropical Atlantic below the ITCZ (Figure 4 and 5d). In the Sargasso Sea area (Figure 5c), simulations and observations consistently show an increase at the end of the year.

The distribution of parameters for the ensemble $\Omega_{100}$ is given in Figure 6. There is a significant dispersion of most of the parameters, suggesting either low sensitivity to the parameters in question (as discussed below) or interdependency between them. The analysis shows that there are very different sets of parameters, within the prescribed ranges, that lead to similar seasonal biomass distribution. (Figure 5). Having said that, we observe that the mortality parameters *m* and $\lambda_m$ minimizing L are biased toward low and high values, respectively. This highlights the key importance of this mortality function in

representing the seasonal distribution. The nitrogen half-saturation constant $K_N$ and the maximum uptake rate of nitrogen $V_{Nmax}$ are also distributed toward low and high values, respectively, suggesting some sensitivity of the *Sargassum* distribution to nitrogen resources. These parameters are poorly constrained by observations and such exercise will allow us to refine the optimization ranges for future studies.



The relative mean deviation of the *Sargassum* seasonal biomass to 10% variations in model parameters shown in Figure 7
confirms findings from the analysis of parameter dispersion in Figure 6. The most influential parameters are the growth rate,
mortality dependence to temperature, and parameter controlling the nitrogen uptake $V_{Nmax}$. It also highlights the influence of
the minimum N-quota ($Q_{Nmin}$). There is moderate dependence on temperature and light, which is in good agreement with
measurements from Hanisak and Samuel (1987). In agreement with previous Lagrangian studies (Berline et al. 2020, Putman
et al. 2020), we find some sensitivity to the windage parameter.

The simulated stranding from the ensemble $\Omega_{100}$ is shown in Figure 8. At the scale of the Tropical Atlantic, these stranding
sum to 1.05 million tons dry weight for the whole year. The predicted strandings in the Caribbean, Northern Brazil, French
Guiana, Sierra Leone, correspond with our current knowledge of *Sargassum* invasions (Bernard et al., 2019; Louime et al.,
2017; Oviatt et al., 2019; Sissini et al., 2017; Oyesiku and Egunyomi 2014, Smetacek and Zingone, 2013). Nevertheless, there
are no available large-scale coastal observations or estimates of stranding to go further in the validation of these simulated
stranding.

A peculiarity of our modelling strategy is to consider the Stokes drift. The Stokes drift induces a displacement of material
parallel to the direction of wave propagation which directly transports the *Sargassum*. An ensemble of 100 simulations without
Stokes drift ("*NoStokes*"), considering the set of physiological parameters from $\Omega_{100}$, has therefore been conducted. Anomalies
of annual *Sargassum* distribution with respect to $\Omega_{100}$ are shown in Figure 9a. *Sargassum* coverage is significantly increased
in the Central Atlantic but decreases sharply in the Caribbean and the Southwestern part of the domain. This highlights the
influence of waves, most probably due to the trade winds, in shaping the seasonal distribution, and transporting the algae
southward.

## 4.2 Sensitivity to external nutrient forcing

We now use the *Sargassum* model to explore how and to which extent continental nutrient sources (riverine nutrient fluxes,
dust deposition, atmospheric N deposition) could participate in the proliferation and may shape the seasonal distribution of
*Sargassum*. First, TATL025BIO simulations were run by deactivating the river sources (simulation "*noriver*"), atmospheric
deposition of P, Si Fe (simulation "*nodust*") or atmospheric deposition of N (simulations "*noNdepo*"). We choose to distinguish
between dust and N deposition because they have not the same origin and because spatial, seasonal and long-term variability
are not necessarily the same.  The simulations were produced from 2006 so the long-term influence of these forcing on the
biogeochemical content is considered there. These simulations were then used to force three ensembles of 100 simulations
which used the sets of parameters from $\Omega_{100}$.

As expected for the "*noriver*" ensemble, the Western Tropical Atlantic, which is under influence of the Amazon and
Orinoco rivers, is experiencing a decrease in nutrient concentrations (Figure 10b). Nutrients also show a decrease in the
Equatorial Atlantic. But for other regions, such as the Sargasso Sea or the Guinea Dome, the long-term equilibration results in
an increase in nitrogen concentration. The resulting *Sargassum* coverage shows a negative anomaly in the Caribbean Sea, in





the Gulf of Mexico and in the region of the ITCZ, with an annual mean basin scale biomass decrease (Figure 9a). For this ensemble, strandings are decreased from 1.05 Mt to 0.78Mt (-25%).

The "*nodust*" ensemble *Sargassum* distribution shows a slight positive anomaly over most of the domain (Figure 9c), particularly in the Central Atlantic and off West Africa, with a slight decrease in the Caribbean Sea and the Gulf of Mexico.

Cumulative strandings sum to 1.1 Mt, and so, are slightly increased compared to the baseline ensemble (+4%). This sensitivity can be explained by the large-scale phosphorus increase and regional increases in nitrogen and phosphorus concentrations (Figure 10) in the biogeochemical simulation. This sensitivity is expected to be produced by the reduced iron concentrations, which limit the phytoplankton growth and thus the nutrient uptake.

The removal of atmospheric nitrogen deposition ("*noNdepo*") leads to a global decrease in surface nitrogen concentration

and an increase in surface phosphorus concentrations across the entire domain (Figure 10). The resulting *Sargassum* coverage is significantly decreased over the whole domain, and the annual averaged stranding decreases by 10% (Figure 9d).

## 5 Discussion and summary

Since 2011, unprecedented massive stranding of the holopelagic *Sargassum* have been reported on the coasts of the Caribbean Sea, Northern Brazil and West Africa. In this paper we developed an Eulerian model of *Sargassum*, which integrates transport,

strandings and algal physiology. The *Sargassum* model is based on the ocean modelling platform NEMO, and is forced by the physical and biogeochemical fields of a regional model (TATL025BIO) as well as by the ERA5 wave and wind fields. An ensemble approach has been used to optimize the physiological parameters. The results demonstrate the ability of the model to represent the spatial distribution and seasonal cycle of the *Sargassum* biomass in the West Atlantic and the Caribbean Sea.

While windage and inertial effects are considered of importance for the drift properties and large scale advection (Brook

et al. 2019, Berline et al. 2020, Beron-Vera et al. 2020, Putman et al. 2018, 2020), we show here that Stokes drift has also significant impacts on the distribution of the *Sargassum*, and in particular on their entrance in the Caribbean Sea. In addition to the anomalous currents that may be at the origin of the *Sargassum* bloom in 2011 (Johns et al. 2020), wave drift could also have contributed to the dissemination of the algae toward the Equatorial Atlantic in the early 2010's. Wave transport of algae is therefore an important component of *Sargassum* modelling that has not yet been accounted for in previous modelling efforts

(Brooks et al., 2018; Putman et al., 2018; Wang et al., 2019, Johns et al. 2020) and should deserve further attention.

The ability of the model to simulate the large-scale distribution was also used to conduct sensitivity tests on the nutrient forcing from rivers, dust and atmospheric deposition. Here, it is worth remembering that the *Sargassum* model is not coupled with the biogeochemical model so it is not directly forced by these external inputs of nutrients but through the biogeochemical model. This prevents the representation of some opportunistic utilization of nutrients that could be done by the algae. With

this limitation in mind, our results evidence that riverine nutrient and atmospheric nutrient inputs could feed the basin scale growth of *Sargassum*, in accordance with various hypothesis in the literature (e.g. Wang et al. 2019). We found a 25% and





10% decrease in annual *Sargassum* distribution in the experiments without river nutrient runoff and without atmospheric nitrogen deposition, respectively. This means that 1) *Sargassum* distribution is sensitive to these forcings, but 2) these forcings alone cannot control the total *Sargassum* biomass. At this stage, the quantification of their role in the interannual variability
and overall increase of Sargassum remains an open question that will deserve further attention.

Several aspects which could be of potential importance for *Sargassum* growth have not been considered here. First, growth and mortality could depend on the age of the fragments, through colonization by epiphytes. There is a lack of knowledge on these aspects and mesocosm experiments would be useful to better constrain such dependence in the model, if relevant. Second, we assume that *Sargassum* does not compete with phytoplankton for resources. How much this hypothesis is valid remains an
open question. The *Sargassum* coverage per area of 0.5×0.5° is rarely above 0.1% (*e.g.* Wang et al. 2019). So, at regional scale, it seems reasonable to consider that *Sargassum* growth does not affect phytoplankton growth. But at the local scale (scale of a raft) and particularly with low mixing conditions, *Sargassum* could compete with phytoplankton for resources. A full coupling of our *Sargassum* model with PISCES-Q may allow us to address such questions. Third, our results show a strong dependence on the nitrogen uptake parameters. We do not consider possible fixation of atmospheric N through diazotrophic
assemblage. Biological nitrogen fixation by diazotrophic macroalgal association has been shown to be important for some *Sargassum* species (e.g. *Sargassum horneri*, Raut et al. 2018) and this could also be the case for the holopelagic *Sargassum* where epibionts N-fixating bacteria have been observed on these species (Carpenter 1972, Michotey et al. 2020).

Finally, our modelling system succeeds in maintaining some biomass in the Central and Eastern Tropical Atlantic. This pool of *Sargassum* has been shown to be of key importance in the year-to-year maintenance of the population (Wang et al.
2019). So, we expect the model to be useful to address the question of interannual variations.

**Acknowledgments**

This study was supported by IRD, CNRS, French *Ministère de la Transition Écologique et Solidaire,* ANR project FORESEA (https://sargassum-foresea.cnrs.fr), and TOSCA SAREDA-DA project. Supercomputing facilities were provided by GENCI project GEN7298.

**Data availability**

The Sargassum model is built upon the standard NEMO code (release 4.0.1, rev 11533). The reference code can be downloaded from the NEMO website (http://www.nemo-ocean.eu/, last access: 11 September 2019). The NEMO code modified to include the Sargassum physiology and transport is available in the Zenodo archive (Jouanno and Benshila 2020, http://doi.org/10.5281/zenodo.4275901). Forcing fields for year 2017 are also included in the Zenodo archive.




## Author Contribution

JJ and RB developed the model code, with inputs from LB, FD, TC, JS, CC on the Sargassum modelling strategy. FD, TT, TC provided insight on the Sargassum physiology. SB, OA, CE and MER participated to the physical-biogeochemical model implementation and tuning. LB provided the Sargassum satellite detection product. PN and MM provided fields of atmospheric
nitrogen deposition. JJ and AS performed the simulations and analysis. JJ prepared the manuscript with contributions from all co-authors.

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





| Parameter | Description | Parameter range | Parameters for baseline simulation | Unit | Reference |
|---|---|---|---|---|---|
| $\mu_{max}$ | Maximum uptake rate of carbon | [0.05-0.09] | 0.069 | $d^{-1}$ | Lapointe et al. (2014) |
| $I_{opt}$ | Optimal light intensity | [60-80] | 72.7 | $W\ m^{-2}$ | Hanisak and Samuel (1987), Lapointe et al. (1995) |
| $K_N$ | Half saturation constant for N uptake (NO3+NH4) | [0.001-0.01] | 0.0014 | $mmol\ m^{-3}$ | This study |
| $K_P$ | Half saturation constant for P uptake (PO4) | [0.001-0.01] | 0.0044 | $mmol\ m^{-3}$ | This study |
| Tmin | Lower temperature limit below which growth ceases, | [10 – 14] | 10.8 | °C | Hanisak and Samuel (1987) |
| Tmax | Upper temperature limit above which growth ceases | [40 – 50] | 43.2 | °C | This study. |
| Topt | Optimum temperature at which growth is maximum | [22-28] | 26.0 | °C | Hanisak and Samuel (1987) |
| m | Maximum mortality rate | [0.05-0.1] | 0.04 | $d^{-1}$ | This study |
| $m_{LC}$ | Maximum sinking rate | [0.05 - 0.1] | 0.0.057 | $d^{-1}$ | This study |
| $\lambda_m$ | Coefficient of the exponential slope for mortality dependance to temperature | [0.2 – 0.7] | 0.62 | | This study |
| $\lambda m_{LC}$ | Coefficient of the exponential slope for Langmuir mortality to depth | [0.2 - 0.7] | 0.28 | | This study |
| $V_{Nmax}$ | Nitrogen maximum uptake rate | [5 $10^{-4}$ – 3 $10^{-3}$] | 1.23 $10^{-3}$ | $mgN\ (mgC)^{-1}\ d^{-1}$ | Lapointe et al. (1995) |
| $V_{Pmax}$ | Phosphorus maximum uptake rate | [9 $10^{-5}$ – 7 $10^{-4}$] | 3.78 $10^{-4}$ | $mgP\ (mgC)^{-1}\ d^{-1}$ | Lapointe et al. (1995) |
| $Q_{Nmin}$ | Minimum N quota | [0.033 – 0.058] | 0.028 | $mgN\ (mgC)^{-1}$ | Lapointe et al. (1995) |
| $Q_{Nmax}$ | Maximum N quota | [0.016 – 0.029] | 0.035 | $mgN\ (mgC)^{-1}$ | Lapointe et al. (1995) |
| $Q_{Pmin}$ | Minimum P quota | [0.0025 – 0.0035] | 0.003 | $mgP\ (mgC)^{-1}$ | Lapointe et al. (1995) |
| $Q_{Pmax}$ | Maximum P quota | [0.005 – 0.0125] | 0.011 | $mgP\ (mgC)^{-1}$ | Lapointe et al. (1995) |
| Windage | Direct wind effect on the *Sargassum* raft displacement | [0 – 1.5] | 1.12 | % | Berline et al. (2020), Putman et al. (2020) |

**Table 1.** *Sargassum* model parameters.



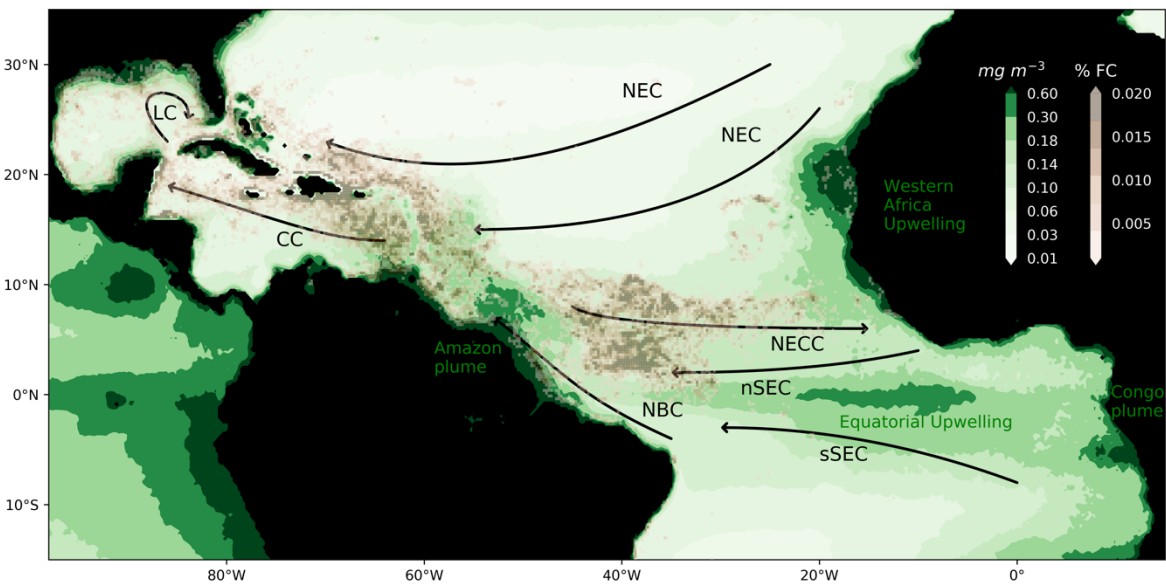

**Figure 1.** *Sargassum* fractional coverage obtained from MODIS in July-August 2017 (Berline et al. 2020, brown color scale; value between 0.001 and 0.02%) and surface chlorophyll distribution in July-August (green color scale; im mg m$^{-3}$) based on GlobColour MODIS monthly product from 2010 to 2018. Circulation schematic of the surface currents is superimposed: the North Equatorial Current (NEC), the northern and southern branches of the South Equatorial Current (nSEC and sSEC), the North Equatorial Countercurrent (NECC), the North Brazil Current (NBC), the Caribbean Current (CC), and the Loop Current 570 (LC).

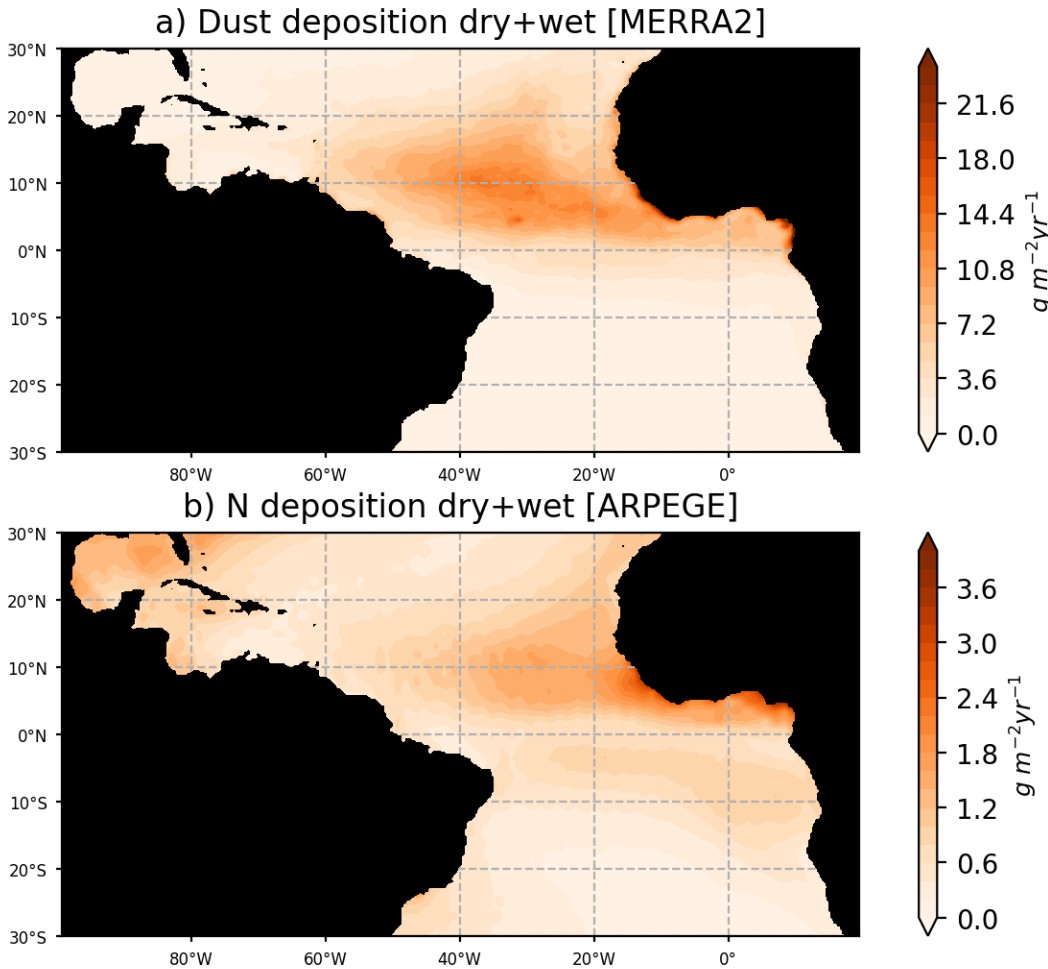

**Figure 2**. Dust fluxes to the ocean from MERRA2 reanalysis for year 2017 (a), and nitrogen flux to the ocean from ARPEGE simulations averaged over the period 2010-2014. Nitrogen fluxes consider the dry and wet fluxes at the ocean surface of $NO_3$, $NH_4$ and $NH_3$. Dust fluxes consider the total dust dry-plus-wet deposition product (DUDPWTSUM) from MERRA2.

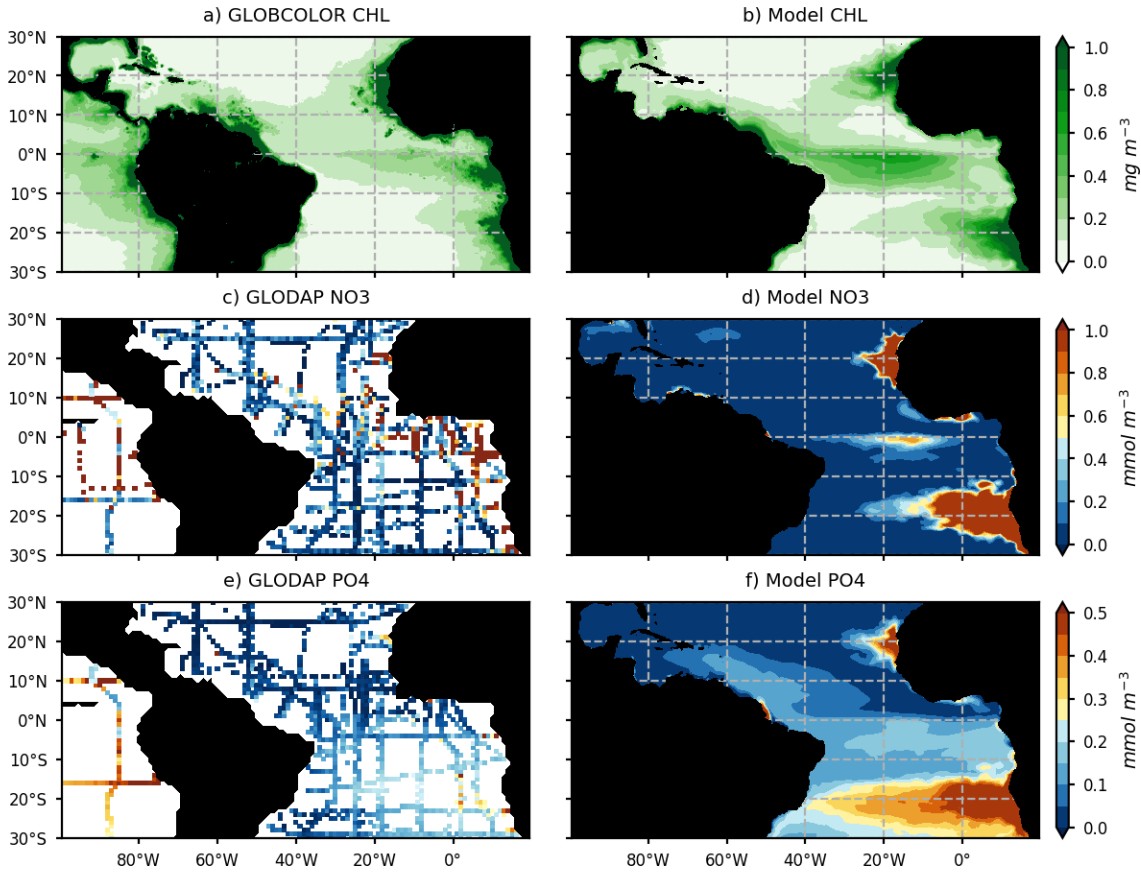

**Figure 3**. Annual mean surface chlorophyll concentrations (mg chl-a m$^{-3}$) from Globcolour satellite product (a) and model (b) for year 2017. Spatial distribution of surface NO$_3$ and PO$_4$ concentration (c,e ; mmol m$^{-3}$) from historical cruises of the GLODAPv2_2016 database (https://www.nodc.noaa.gov/ocads/oceans/GLODAPv2/) and annual mean surface NO$_3$ and PO$_4$ distribution from the model (d,f).



**Figure 4.** Seasonal distribution of *Sargassum* Fractional Coverage for year 2017 from observations (left) and from a selected ensemble of 100 simulations ($\Omega_{100}$) with varying parameters (right).



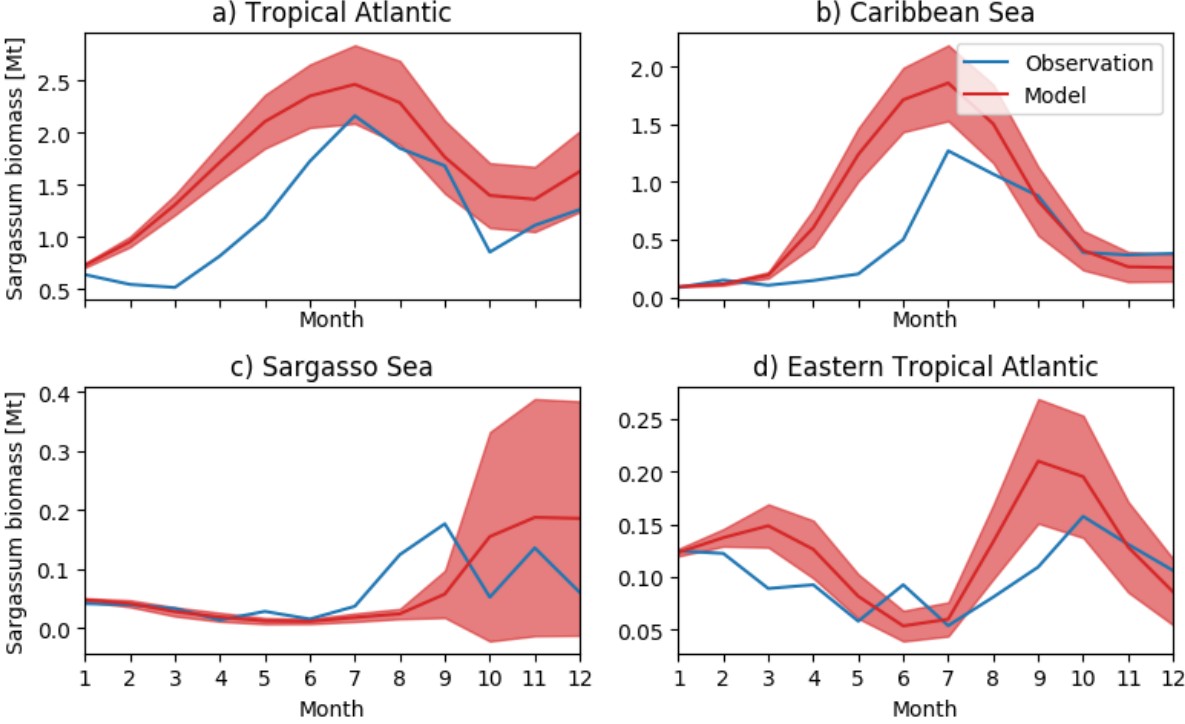

**Figure 5.** Seasonal distribution of *Sargassum* Fractional Coverage for year 2017 from observations (blue) and from a selected ensemble of 100 simulations ($\Omega_{100}$) with varying parameters (mean in red and variance in shaded red) averaged in different areas: (a) Tropical Atlantic [98°W-10°W;0°N-30°N], (b) Caribbean Sea [85°W-55°W;8°N-22°N], (c) Sargasso Sea [80°W-50°W;23°N-30°N], and Eastern North Tropical Atlantic [30°W-0°E;0°N-15°N].



**Figure 6.** Likelihood (L) as a function of the different parameters that have been varied in the optimization experiment, shown here for the ensemble $\Omega_{100}$ that composes the ensemble averages in Figures 4 and 5. Units are given in Table 1.



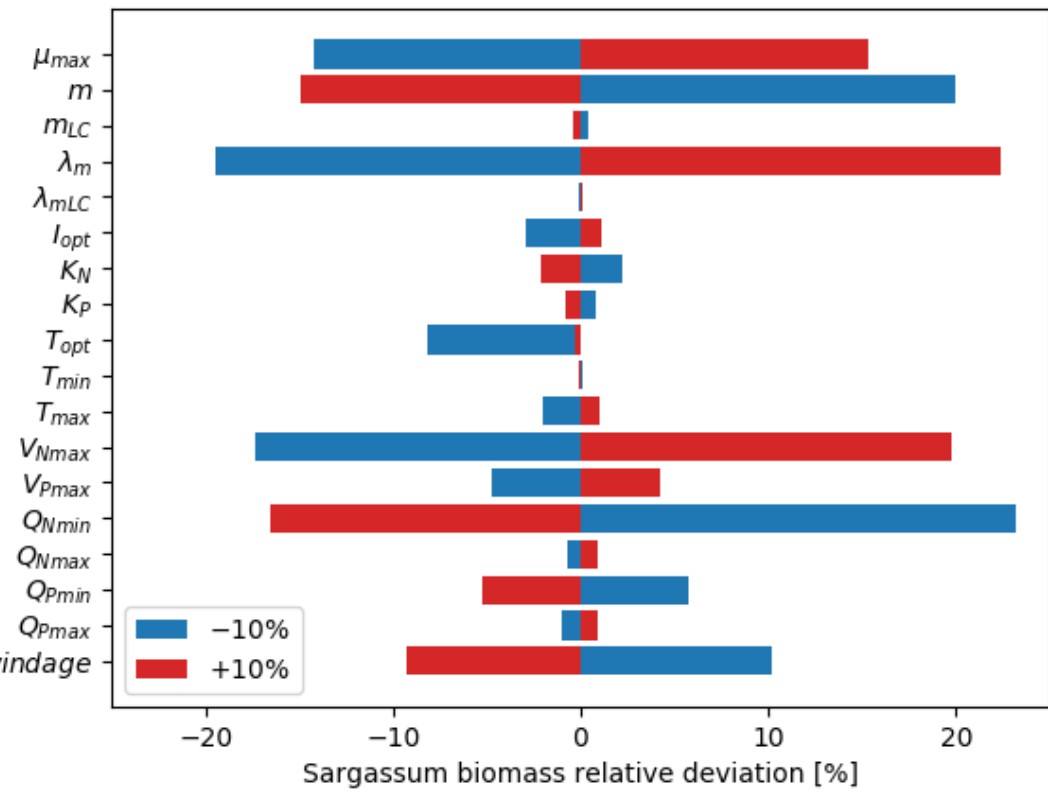

595

**Figure 7**. Sensitivity analysis to model parameters expressed as the mean relative deviation (%) between the baseline simulation and the simulation in which one parameter was modified by ±10%. The set of parameters for the baseline simulation are given in Table 1.





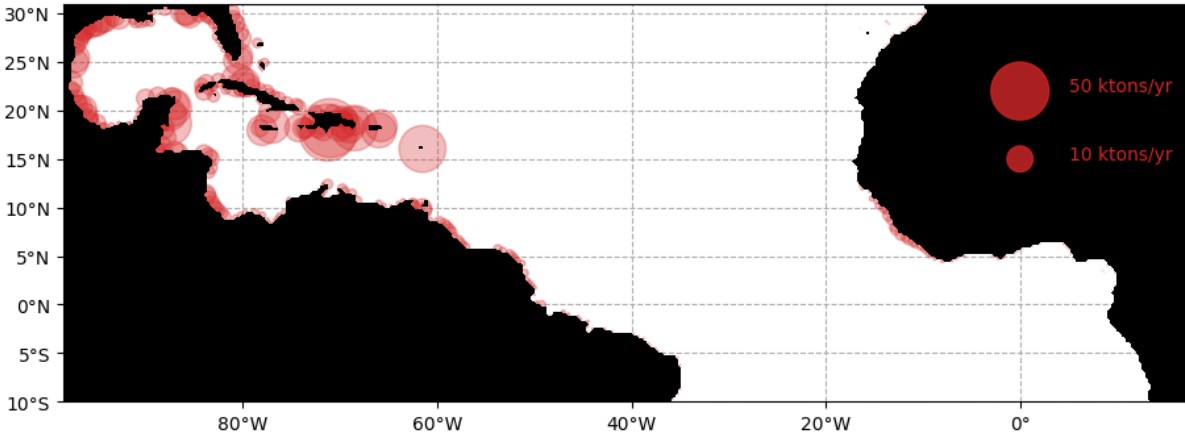

600

**Figure 8.** Cumulated annual *Sargassum* wet biomass stranding per area of 25x25km for year 2017 obtained from the ensemble $\Omega_{100}$.





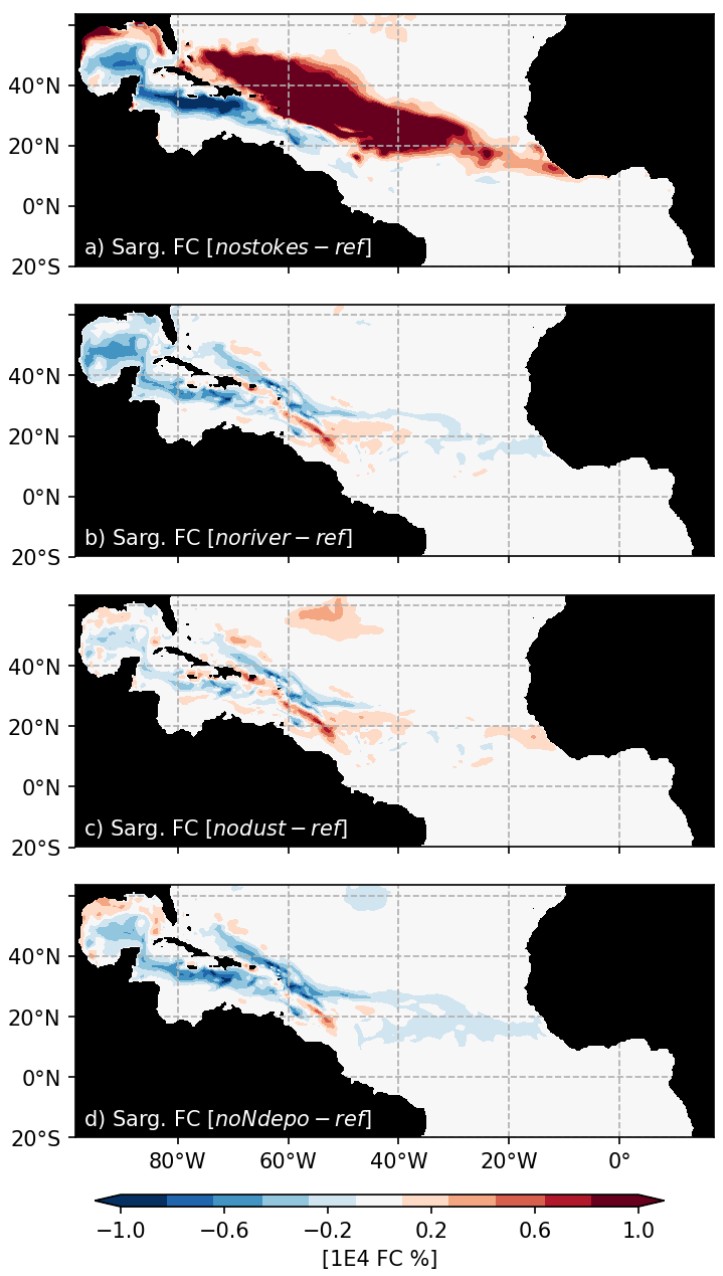

**Fig. 9.** Ensemble anomalies of *Sargassum* coverage (in % of surface) from sensitivity ensemble simulations of 100 members each, which were performed with the set of parameters from $\Omega_{100}$.



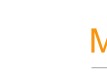
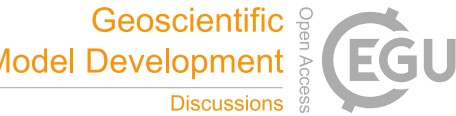

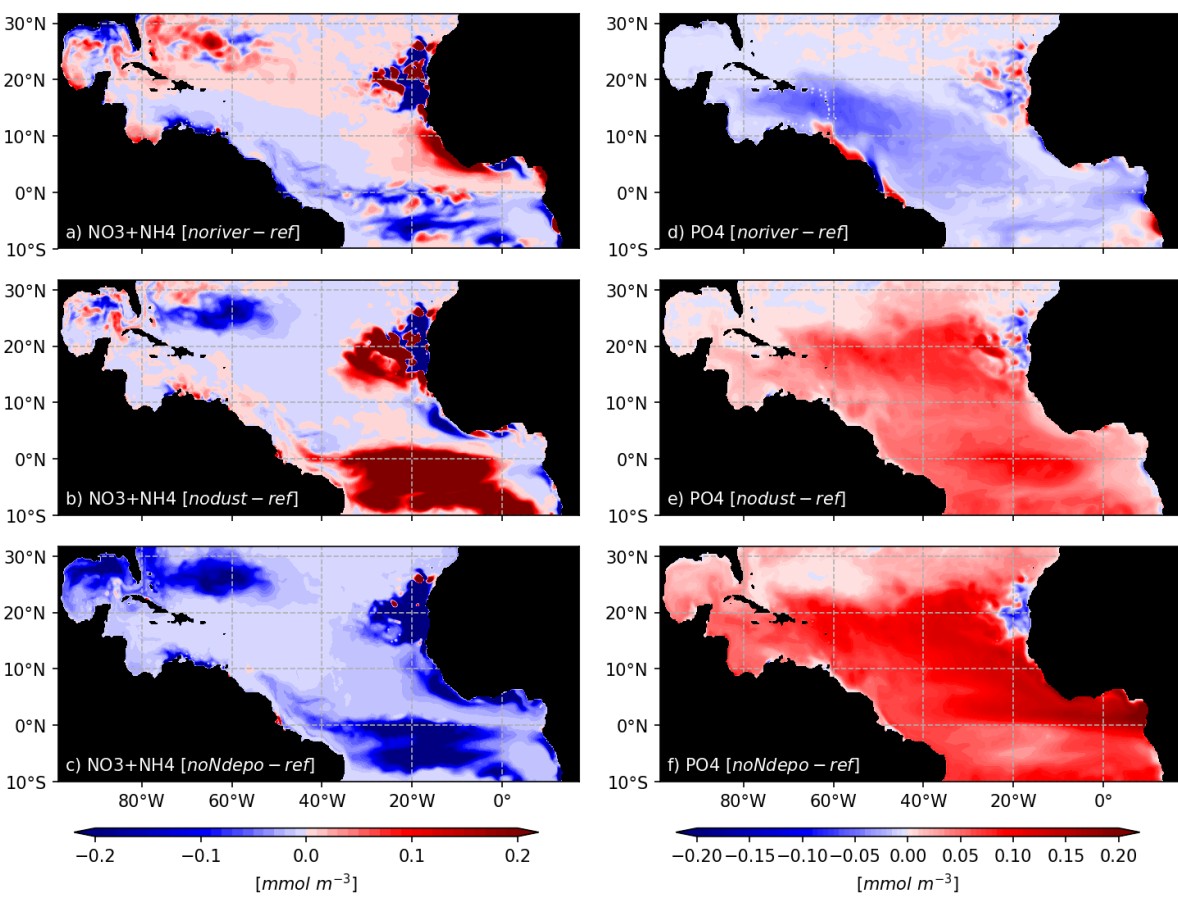

**Fig. 10.** Biogeochemical response to sensitivity experiments to river, dust and N deposition: surface anomalies of $NO_3 + NH_4$ (left) and $PO_4$ (right) with respect to the reference TATL025BIO simulation.