# Peer review of "A NEMO-based model of *Sargassum* distribution in the Tropical Atlantic: description of the model and sensitivity analysis (NEMO-Sarg1.0)"

_Geoscientific Model Development, 2020_

## Referee Comment (RC1) · Anonymous Referee #1 · 5 Jan 2021

This paper proposes a first attempt at 2D biogeochemical modelling of an important present example of pelagic algal mass proliferation, due to the Sargassum genus in the tropical Atlantic area. As mass strandings of these floating Pheophyceae are very harmful for the Carribean economy and, potentially, for human populations, trying to deterministically explain this phenomenon and to assess the respective roles of several potential causes appears to be fully required. By using a deterministic modelling approach, the authors have taken the right path, but some questions remain unsolved in this first model. 1/ Why is the Sargassum module not embedded in the global biogeochemical model? The present model is a 2D layer of surface water (thickness = 1m), in which the nutrient pools (as well as the concentrations of the various phytoplanktonic competitors of the macrophytes) are forced daily from a pre-existing model of pelagic ecosystem in which Sargassum was absent! So, nutrient depletion is not induced by Sargassum mass proliferation and Sargassum cannot win any competition with phytoplankton. In classical eutrophication models, a feedback from the growing macrophytes towards dissolved nutrients is considered as a key control of the process. In order to make their assumption of no feed-back more acceptable, the authors should at least provide in their discussion a quantitative estimation of the daily consumption of inorganic nutrients N and P by Sargassum mean biomass and compare it to the phytoplanktonic consumption and the existing dissolved stock. If the Sargassum uptake should appear to be of same order of the phytoplankton one, the model should be re-run with the Sargassum module included in the biogeochemical model. 2/ The simulated Sargassum compares favorably with satellite observations in July-October, but shows heavy proliferations in March-June which seem not to be observed. Why ? Is the temperature the main driver for that ? 3/ The most interesting (and "hot") question lies in the role of recently increased river inputs of nutrients. The paper should show more clearly the extension of the main river plumes, not in terms of salinity, but perhaps in terms of %of increase of ambient natural nutrients. Moreover, numerous eutrophication models (Radtke et al., 2012; . Troost et al., 2013; . Dulière et al., 2017; Große et al., 2017; Lenhart and Große, 2018) have now used the numerical tracer method initially proposed by Ménesguen et al. (2006) to track in the whole biogeochemical net the nitrogen or the phosphorus coming from any source. Application to the Amazon could for instance quantify its effective role in the Sargassum mass proliferation. 4/ As indicated lines 144-145, this paper is not the first one taking into account the drift of macroalgae in eutrophication modelling. See Bergamasco and Zago (1999), Brush and Nixon (2010) or Ménesguen et al. (2006), 5/ Some Sargassum parameters must be more precisely founded • The values of the half saturation constant for N uptake (NO3+NH4) (0.0014 mmol m-3) and the half saturation constant for P uptake (0.0044 mmol m-3) seem to be very very low, so that the ambient nutrient increase by river loadings can be expected to exert no influence on the Sargassum nutrient limitation. The

authors should find measured values in the literature, and compare their half saturation constants to the mean ambient concentrations of dissolved nutrients.  c Maximum sinking rate is not correctly defined in Table 1 : 0.0.057 (probably 0.057 d-1)

References Bergamasco, A., Zago, C., 1999. Exploring the nitrogen cycle and macroalgae dynamics in the lagoon of Venice using a multibox model. Estuar. Coast. Shelf Sci. 48, 155–175. Brush, M.J., Nixon, S.W., 2010. Modeling the role of macroalgae in a shallow sub-estuary of Narragansett Bay, RI (USA). Ecol. Model. 221, 1065–1079. V. Dulière, N. Gypens, C. Lancelot, P. Luyten, G. Lacroix, 2017. Origin of nitrogen in the English Channel and Southern Bight of the North Sea ecosystems. Hydrobiologia, 845, 13–33. F. Große, M. Kreus, H.-J. Lenhart, J. Pätsch, Pohlmann T., 2017. A novel modeling approach to quantify the influence of nitrogen inputs on the oxygen dynamics of the North Sea. Front. Mar. Sci., 4, 1-21. H.J. Lenhart, F. Große, 2018. Assessing the effects of WFD nutrient reductions within an OSPAR frame using trans-boundary nutrient modeling. Front. Mar. Sci., 5, 447 A. Ménesguen, P. Cugier, I. Leblond, 2006. A new numerical technique for tracking chemical species in a multisource, coastal ecosystem, applied to nitrogen causing Ulva blooms in the Bay of Brest (France). Limnol. Oceanogr., 51 (1, part 2), 591-601. H. Radtke, T. Neumann, M. Voss, W. Fennel, 2012. Modeling pathways of riverine nitrogen and phosphorus in the Baltic Sea. J. Geophys. Res., 117 T.A. Troost, M. Blaas, F.J. Los, 2013. The role of atmospheric deposition in the eutrophication of the North Sea: A model analysis. Journal of Marine Systems,125, 101-112,

---

## Referee Comment (RC2) · Anonymous Referee #2 · 1 Mar 2021

This is a very useful and valuable paper that will help contribute to the understanding of Sargassum. I recommend it for publication following the authors' consideration of the below points.

First sentence abstract: Consider altering tense "The Tropical Atlantic has been facing...."

Second sentence abstract: Consider being more specific about what kind of "Sargassum modelling" you mean. Transport? Growth? Population dynamics? Life-cycle? Food-web? You mention specifically what you do shortly thereafter, but I think some clarifying what type of model is needed for seasonal forecasts would be helpful.

First sentence introduction: Consider rewording to "... in the Northern Tropical Atlantic Ocean from 2011 to present causes annual..."

Line 37: consider a different word from "evolution." Many of your readers are biologists and will think that you are referring to genetic change through time (e.g., via natural selection) and I don't think that is how you intend the word to be used.

Line 71: what do you mean by "wind, wave or any event" ? Do you mean any weather event?

Line 99: I am not an expert on ocean circulation models, but it has been shown in some cases that 1/4 degree resolution models can "average out" sometime important aspects of ocean circulation. How does this relatively coarse grid resolution bias our view of the importance of wind/waves? Are these results model configuration specific? See: Putman, N.F. and He, R., 2013. Tracking the long-distance dispersal of marine organisms: sensitivity to ocean model resolution. Journal of the Royal Society Interface, 10(81), p.20120979.

Line 113: Daily fields are used for the PICES model, what did you use for the NEMO-based ocean circulation model? Also daily?

General Methods: can you clarify details on the wind/wave models(data?) used for testing the influence of windage and Stokes drift?

Line 288: Is this an ocean model resolution issue? Perhaps the energetic eddy fields are not well resolved and the simulated Sargassum moves more linearly into the Caribbean?

Table 1: I am a little confused on the Parameter range for Windage. The Putman et al. paper cited tracked Sargassum mats with GPS devices and compared movements to predictions in the operational hindcast HYCOM model and used winds from NOAA's Blended Sea Winds. Their simulations that best matched the actual Sargassum mats used Windage values of 1% or 3%. They noted that the particular value was likely

[Figure]

dependent upon the ocean and wind models. I am not asking you to conduct new simulations, but can you explain why you chose not to consider stronger values of windage in your simulations? Figure 8: This is a very interesting Figure. There is certainly good agreement on the Sargassum beaching locations in Africa and the Caribbean. However, I also find it interesting how much beaching is predicted in the Gulf of Mexico. My understanding is that there has been very little stranding in the western Gulf of Mexico over the past several years (I believe including 2017). Can you explain this discrepancy if it is a "false positive"?

---

## Author Comment (AC1) · 16 Apr 2021

**Reply to comments on "A NEMO-based model of Sargassum distribution in the Tropical Atlantic: description of the model and sensitivity analysis (NEMO-Sarg1.0)" by Jouanno et al.**
**11 Jan 2020**

We thank the reviewers for their time, their careful reading and their insightful comments. We do think they offer very interesting perspectives on our work and its future development.

Reviewer #1 mainly raised issues on the physiological/biogeochemical modelling aspects of our work. We thank the reviewer for having made known to us important references on macroalgae modelling which are now included in the manuscript and the discussion. Following the reviewer's request, we verified that, indeed, N and P consumption by Sargassum is much weaker than consumption by the phytoplankton. This and other reasons given in the detailed response below, reinforce our choice to not embed the sargassum module in the biogeochemical model PISCES. The use of a nutrient tracking procedure, initially developed by Ménesguen et al. (2006) offer a very interesting perspective for tracking the influence of the different nutrient inputs on Sargassum growth (and more generally on the phytoplankton distribution). But the application of these ideas in the biogeochemical model and in the Sargassum model require important developments in the code that we feel are too substantial at this stage and would deserve a dedicated study. As noted by reviewer #1, this is an initial (but comprehensive) modelling effort that we believe provides interesting results that hopefully will gain in accuracy/realism in the near future.

Reviewer #2 mainly raised issues on the physics and modelling of Sargassum transport. Modelling the surface transport, and its dependence on model resolution and/or the level of coupling at the air-sea interface (current/wave/wind interactions) is clearly a hot question for the oceanographic community. To the best of our knowledge, no definitive answers to some of the questions/issues raised by reviewer #2 are currently available, but we have tried to give some hints on possible answers as well as limitations of our study. Important here is the fact that there are observations to gauge the model results.

Finally, we would like to mention that results reported in the new manuscript use a modified irradiance limitation function in the model. In the first version of the manuscript, we used the irradiance limitation function from Martins and Marques (2002) for Enteromorpha sp. We choose here to modify this function in order to mimic results by Hanisak and Samuel (1987) that did not reveal growth limitation at high irradiance conditions for Sargassum. This change slightly improved the overall results of the simulations, and provided a new set of parameters for our baseline simulation. Last but not least: all our conclusions are confirmed by this new set of parameters.

Julien Jouanno on behalf of the authors.

**REFEREE 1**

This paper proposes a first attempt at 2D biogeochemical modelling of an important present example of pelagic algal mass proliferation, due to the Sargassum genus in the tropical Atlantic area. As mass strandings of these floating Pheophyceae are very harmful for the Carribean economy and, potentially, for human populations, trying to deterministically explain this phenomenon and to assess the respective roles of several potential causes appears to be fully required. By using a deterministic modelling approach, the authors have taken the right path, but some questions remain unsolved in this first model.

1/ Why is the Sargassum module not embedded in the global bio- geochemical model? The present model is a 2D layer of surface water (thickness = 1m), in which the nutrient pools (as well as the concentrations of the various phyto- planktonic competitors of the macrophytes) are forced daily from a pre-existing model of pelagic ecosystem in which Sargassum was absent! So, nutrient depletion is not induced by Sargassum mass proliferation and Sargassum cannot win any competition with phytoplankton. In classical eutrophication models, a feedback from the growing macrophytes towards dissolved nutrients is considered as a key control of the process. In order to make their assumption of no feed-back more acceptable, the authors should at least provide in their discussion a quantitative estimation of the daily consumption of inorganic nutrients N and P by Sargassum mean biomass and compare it to the phytoplanktonic consumption and the existing dissolved stock. If the Sargassum up-take should appear to be of same order of the phytoplankton one, the model should be re-run with the Sargassum module included in the biogeochemical model.

There are several reasons why we choose not to embed the sargassum module in the biogeochemical model. First, there is so much uncertainty about the growth/mortality properties of *Sargassum Sp* and in the parameters for its modelling, that we gave priority to the possibility of producing a large ensemble of simulations (more than 10000) to investigate the sensitivity of our results. Due to the high computational cost of the biogeochemical model, this was only possible if the sargassum model was decoupled. Second, such strategy offers some versatility since, we hope, this separate Sargassum module can be readily used by other modeling groups. Third, results shown in the manuscript confirm the capacity of the model to reproduce the sargassum seasonal distribution despite the no feed-back assumption. And fourth, following your recommendation, we compared the N and P uptake by the Sargassum (Figure S1ab) and by the phytoplankton (Figure S1cd). The consumption of N and P by the phytoplankton is two to three orders of magnitude larger than the consumption of N and P by the Sargassum, justifying the decoupling. Moreover, it is worth noting that that high consumption regions for Sargassum do not coincide with those of phytoplankton.

We added figure S1 to the manuscript (as Figure 11) and address these issues/results in the discussion section.

Overall, the results suggest (fortunately!) there is no eutrophication issue with the Sargassum proliferation in the open ocean. However, at the scale of a bay or of a lagoon where Sargassum tend to accumulate, their interactions with the local biogeochemical cycles would definitely need to be addressed. But the analysis of such local scales are out of the scope of this study.

2/ The simulated Sargassum compares favorably with satellite observations in July-October, but shows heavy proliferations in March-June which seem not to be observed. Why ? Is the temperature the main driver for that ?

Temperature is well constrained in the region in our NEMO simulations (mean bias is usually below 0.5°C; e.g., see Hernandez et al. 2016) and temperature biases in the western tropical Atlantic use to be at their lowest weaker in spring. We expect much larger bias in the simulated nutrients and also in the Sargassum biomass estimate from space (this area is very cloudy with contrasted aggregative properties). The sargassum physiology and parametrization choices are poorly constrained by observations and could also be an important source of error. But at this point, it is really hard to conclude on the causes of these differences.

*Hernandez, O. , J. Jouanno and F. Durand (2016). Do the Amazon and Orinoco river plumes influence tropical cyclone-induced surface cooling. Journal of Geophysical Research: Oceans, 121(4), 2119-2141.*

[Figure]

*Figure S1: Average N and P uptake by phytoplankton (in mmol m⁻² d⁻¹; subfigures (a) and (b), respectively) and by Sargassum (in μmol m⁻² d⁻¹; subfigures (c) and (d), respectively). The N and P uptake by phytoplankton were obtained from the biogeochemical simulation, assuming a constant stoichiometry. The uptake rates were integrated over the model mixed-layer depth for each month of 2017 and averaged over the year. The bottom row shows the Sargassum vs phytoplankton mean consumption ratio of (e) N and (f) P (in ‰) for year 2017.*

3/ The most interesting (and "hot") question lies in the role of recently increased river inputs of nutrients. The paper should show more clearly the extension of the main river plumes, not in terms of salinity, but perhaps in terms of %of increase of ambient natural nutrients. Moreover, numerous eutrophication models (Radtke et al., 2012; Troost et al., 2013; . Dulière et al., 2017; Große et al., 2017; Lenhart and Große, 2018) have now used the numerical tracer method initially proposed by Ménesguen et al. (2006) to track in the whole biogeochemical net the nitrogen or the phosphorus coming from any source. Application to the Amazon could for instance quantify its effective role in the Sargassum mass proliferation.

The impact of the rivers on the nutrient distribution is shown in Figs. 10a,b which compare the surface nutrient distributions in simulations with and without rivers. We can see a reduction of

the surface nutrient (N and P) in the river plume area. One caveat of our sensitivity study is that the Amazon plume is not well resolved by the model (as by many other biogeochemical models), suggesting we are missing some important biogeochemical processes in the river plume. We are currently working on these questions (refinement of the hydro-biogeochemical riverine inputs, photodegradation of the organic material, diazotrophy), but we (and the community) still have a long way to go to correctly represent the large tropical river plumes in basin scales models.

We recognize that the numerical tracing method of Menesguen et al. (2006) would certainly be appropriate to address this issue. But again, it would be affected by biases in the modelled biogeochemical content of the plume. Furthermore, this would require significant developments in the PISCES code (as well as the implementation of a full coupling between PISCES and NEMO-Sarg) that are beyond the scope of this study.

In a recent paper, we addressed specifically the issue of the role of the rivers on the proliferation (Jouanno et al. 2021). Our conclusion, and the order of magnitude of the Amazon River's contribution to proliferation obtained from this modeling work, are in very good agreement with our findings obtained on an observational basis.

*Jouanno, J., J. S. Moquet, L. Berline, M.H. Radenac, W. Santini, T. Changeux, T. Thibaut, W. Podlejski, F. Menard, J.M. Martínez, O, Aumont, J. Sheinbaum, N. Filizola, G.D. Mounkandi N'kaya. Evolution of the riverine nutrient export to the Tropical Atlantic over the last 15 years: is there a link with Sargassum proliferation? Environmental Research Letters, 16(3), 034042.*

4/ As indicated lines 144-145, this paper is not the first one taking into account the drift of macroalgae in eutrophication modelling. See Bergamasco and Zago (1999), Brush and Nixon (2010) or Ménesguen et al. (2006),

Thanks for pointing out these references we missed. We added them to the manuscript. As previously mentioned, the aim of our study is to represent the Sargassum biomass at a basin scale. While these 3 previous studies are considered as pioneering works for our subject, they focused on much finer modelling scales (Bergamasco and Zago (1999) worked at the local scale of a municipality or lagoon, Brush and Nixon (2010) at a sub-estuary scale and Ménesgen et al (2006) at the scale of the Bay of Brest). The difficulty of Sargassum modelling is its large-scale cover.

5/ Some Sargassum parameters must be more precisely founded. The values of the half saturation constant for N uptake (NO3+NH4) (0.0014 mmol m-3) and the half saturation constant for P uptake (0.0044 mmol m-3) seem to be very very low, so that the ambient nutrient increase by river load- ings can be expected to exert no influence on the Sargassum nutrient limitation. The authors should find measured values in the literature, and compare their half saturation constants to the mean ambient concentrations of dissolved nutrients. Maximum sinking rate is not correctly defined in Table 1 : 0.0.057 (probably 0.057 d-1)

Indeed, the N and P half saturation value are rather low. This N/P half saturation constant are the results of our basin scale optimization procedure and they are likely biased to be low because the biogeochemical model tends to have low surface nutrient concentrations in the northern Tropical Atlantic. But it is worth noting that we have no observational results to compare with. We could only find in the literature information of the stoichiometry, the growth rate, the max- imum uptake rates, the sensitivity to temperature and irradiance (see Table 1). To our

knowledge, there is currently no published observations from which we could infer a half saturation constant. The sinking rate value has been corrected, thanks.

References

Bergamasco, A., Zago, C., 1999. Exploring the nitrogen cycle and macroalgae dynamics in the lagoon of Venice using a multibox model. Estuar. Coast. Shelf Sci. 48, 155–175.

Brush, M.J., Nixon, S.W., 2010. Modeling the role of macroal- gae in a shallow sub-estuary of Narragansett Bay, RI (USA). Ecol. Model. 221, 1065– 1079.

V. Dulière, N. Gypens, C. Lancelot, P. Luyten, G. Lacroix, 2017. Origin of nitrogen in the English Channel and Southern Bight of the North Sea ecosystems. Hy- drobiologia, 845, 13–33. F.

Große, M. Kreus, H.-J. Lenhart, J. Pätsch, Pohlmann T., 2017. A novel modeling approach to quantify the influence of nitrogen inputs on the oxygen dynamics of the North Sea. Front. Mar. Sci., 4, 1-21.

H.J. Lenhart, F. Große, 2018. Assessing the effects of WFD nutrient reductions within an OSPAR frame using trans-boundary nutrient modeling. Front. Mar. Sci., 5, 447

A. Ménesguen, P. Cugier, I. Leblond, 2006. A new numerical technique for tracking chemical species in a mul-tisource, coastal ecosystem, applied to nitrogen causing Ulva blooms in the Bay of Brest (France). Limnol. Oceanogr., 51 (1, part 2), 591-601.

H. Radtke, T. Neumann, M. Voss, W. Fennel, 2012. Modeling pathways of riverine nitrogen and phosphorus in the Baltic Sea. J. Geophys. Res., 117

T.A. Troost, M. Blaas, F.J. Los, 2013. The role of atmospheric deposition in the eutrophication of the North Sea: A model analysis. Journal of Marine Systems,125, 101-112.

**REFEREE 2**

This is a very useful and valuable paper that will help contribute to the understanding of Sargassum. I recommend it for publication following the authors' consideration of the below points.

First sentence abstract: Consider altering tense "The Tropical Atlantic has been facing. . .."

Thanks, we followed your recommendation.

Second sentence abstract: Consider being more specific about what kind of "Sargassum modelling" you mean. Transport? Growth? Population dynamics? Life-cycle? Food-web? You mention specifically what you do shortly thereafter, but I think some clarifying what type of model is needed for seasonal forecasts would be helpful.

We modified the sentence as follows: "The development of large-scale modelling of Sargassum transport and physiology is essential to clarify the link between Sargassum distribution and environmental conditions, and to lay the groundwork for a seasonal forecast at the scale of the Tropical Atlantic basin."

First sentence introduction: Consider rewording to ". . . in the Northern Tropical Atlantic Ocean from 2011 to present causes annual. . ."

Thanks for your suggestion. Done.

Line 37: consider a different word from "evolution." Many of your readers are biologists and will think that you are referring to genetic change through time (e.g., via natural selection) and I don't think that is how you intend the word to be used.

We removed "evolution". Finally, all what we meant is included in the term "distribution".

Line 71: what do you mean by "wind, wave or any event" ? Do you mean any weather event?

Here, our point was to mention that rafts can collapse if the dynamical conditions change. We rephrased it as follows: "Individuals in these aggregations can be easily dispersed when the dynamical conditions favorable to aggregation cease (Ody et al., 2019)."

Line 99: I am not an expert on ocean circulation models, but it has been shown in some cases that 1/4 degree resolution models can "average out" sometime important aspects of ocean circulation. How does this relatively coarse grid resolution bias our view of the importance of wind/waves? Are these results model configuration specific? See: Putman, N.F. and He, R., 2013. Tracking the long-distance dispersal of marine or- ganisms: sensitivity to ocean model resolution. Journal of the Royal Society Interface, 10(81), p.20120979.

As raised by Putman and He (2013), there may be issues regarding model resolution. We are working at an intermediate resolution (~ eddy permitting), so we lack some energy at the mesoscale. Since this mesoscale is particularly important for the dynamics in the Caribbean, Gulf of Mexico, or the North Brazil current area, we would expect more realistic transport properties at higher resolution. But our experience is that ¼° NEMO simulations works well in the region on many aspects of the regional dynamics, such as river plume extent (Hernandez et al. 2016, 2017), large scale currents (Kounta et al. 2018), biogeochemistry (Radenac et al. 2020), salinity large scale distribution (Awo et al. 2018) among other. One reason is that the scales of variability in the tropics are larger than at midlatitudes. This is confirmed by the present study since we show that the simulated ocean dynamics are good enough to represent the accumulation of Sargassum in the ITCZ, the advection in the Caribbean through the Antilles, and the episodic shedding of Loop Current eddies in the Gulf of Mexico.

We also expect model resolution to be just one part of the story regarding the dependence of the transport properties to numerics. Surface transport also depends on the vertical resolution of the model in the mixed-layer, the vertical mixing scheme, the degree of coupling of the ocean circulation with the atmosphere or the waves, the wind product used to force the model, etc…

In our model, the windage coefficient and transport act as an empirical factor that compensates lacking the explicit simulation of some of these processes and probably helps us to properly simulate a realistic large scale Sargassum advection.

The physics of how Sargassum is transported is not well understood (see Beron-Vera, F.J. & Miron, P., 2020. A minimal Maxey–Riley model for the drift of Sargassum rafts. Journal of Fluid Mechanics, 904, p.1157) and therefore, the problem goes beyond issues of model resolution, so the success of the models need to be based on how well they compare to the available

observations (however limited) at the scales of variability they intend to simulate. We think that as an initial attempt at tackling this complicated problem our model is sound and robust, but with obvious limitations. Overall, we definitely need to rely on dedicated Lagrangian studies such as the one performed by Putman et al. 2018, Putman and He 2013, Berline et al. 2020, Putman et al. 2020 to better constrain our model, and learn about best practices in terms of forcing sargassum transport.

*Gévaudan, M.,  J. Jouanno, F. Durand, G. Morvan; L. Renault, and G. Samson. Influence of ocean salinity stratification on the tropical Atlantic Ocean climate. Clim Dynamics.*

*Radenac, M.-H., Jouanno, J., Tchamabi, C. C., Awo, M., Bourlès, B., Arnault, S., and Aumont, O. (2020). Physical drivers of the nitrate seasonal variability in the Atlantic cold tongue. Biogeosciences, 17, 529–545.*

*Giffard, P., W. Llovel, J. Jouanno, G. Morvan, and B. Decharme (2019). Contribution of the Amazon River Discharge to Regional Sea Level in the Tropical Atlantic Ocean. Water, 11, 2348.*

*Awo F. M., G. Alory, C. Y. Da-Allada, T. Delcroix, J. Jouanno, E. Baloïtch (2018). Sea Surface Salinity signature of the tropical Atlantic interannual climatic modes. Journal of Geophysical Research, https://doi.org/10.1029/2018JC013837*

*Kounta, L., Capet, X., Jouanno, J., Kolodziejczyk, N., Sow, B., and Gaye, A. T.: A model perspective on the dynamics of the shadow zone of the eastern tropical North Atlantic. Part 1: the poleward slope currents along West Africa, Ocean Sci. Discuss., https://doi.org/10.5194/os-2018-16*

Line 113: Daily fields are used for the PICES model, what did you use for the NEMO- based ocean circulation model? Also daily?

Also daily. This is now mentioned as follows: "The model is run from 2006 to 2017 and daily physical and biogeochemical fields are extracted to force the Sargassum model."

General Methods: can you clarify details on the wind/wave models(data?) used for testing the influence of windage and Stokes drift?

This is now indicated in section "3.3.1 Initialization and forcing" as follows: "The transport is forced by daily velocities from TATL025BIO simulations (see Section 3.1). The windage is forced with 3-hour winds from the DFS5.2 dataset (Dussin et al. 2016) which were used to force the physical model. The stokes drift in the surface layer is computed according to Breivik et al. (2014) and forced with hourly ERA5 stokes drift product."

Breivik, Ø., Janssen, P. A., & Bidlot, J. R. (2014). Approximate Stokes drift profiles in deep water. Journal of Physical Oceanography, 44(9), 2433-2445.

Line 288: Is this an ocean model resolution issue? Perhaps the energetic eddy fields are not well resolved and the simulated Sargassum moves more linearly into the Caribbean?

At ¼° the dynamics in the Caribbean lacks energy but is not linear (see daily snapshot of sargassum distribution for day 17 July 2017 in the baseline simulation below; it shows the presence

of mesoscale features in the Caribbean). The fact that the northern fraction of the Caribbean is more abundantly covered by Sargassum is also observed (Fig 4 left) and is in agreement with northward surface transport in response to the trade winds and large anticyclonic detaching from the southern Caribbean coast (both contributing to surface current divergence in the southern Caribbean and thus contributing to upwelling, see e.g. Jouanno and Sheinbaum 2013)

*Jouanno, J. and J. Sheinbaum (2013). Eddies and heat balance in the Caribbean Upwelling System, Journal of Physical Oceanography, doi :10.1175/JPO-D-12-0140.1.*

[Figure]

Table 1: I am a little confused on the Parameter range for Windage. The Putman et al. paper cited tracked Sargassum mats with GPS devices and compared movements to predictions in the operational hindcast HYCOM model and used winds from NOAA's Blended Sea Winds. Their simulations that best matched the actual Sargassum mats used Windage values of 1% or 3%. They noted that the particular value was likely dependent upon the ocean and wind models. I am not asking you to conduct new simulations, but can you explain why you chose not to consider stronger values of windage in your simulations?

Simply because we also consider stokes drift. As mentioned in Putman et al. 2020 "this windage can reflect the combined effects of direct wind forcing [and] downwind Stokes drift".

Figure 8: This is a very interesting Figure. There is certainly good agreement on the Sargassum beaching locations in Africa and the Caribbean. However, I also find it interesting how much beaching is predicted in the Gulf of Mexico. My understanding is that there has been very little stranding in the western Gulf of Mexico over the past several years (I believe including 2017). Can you explain this discrepancy if it is a "false positive"?

We have no definitive answer regarding this discrepancy. We may have too many false positive detections in the satellite product which contaminates the initialization field (Figure 4 in JF) so we maintain some population there. But it is worth noting that we miss in-situ observations and reporting to confirm/infirm the importance of the strandings in this area.

---

## Author Response (AR2)

Dear Author,

Thank you for your revised version of the manuscript. I consider that your answers to the referees are well argued and this is why I do not ask for a second round of reviews. However, I think that your paper would benefit from adding, in your manuscript, some precisions that you develop in your answers.

Regarding the question of why the Sargassum module is not embedded in the global model, your justification is convincing and you added a new sentence about this in your manuscript, which is fine.

But concerning the heavy proliferations in March-June in your simulations that do not seem to be observed (as noted by Referee #1), please discuss this providing some additional details in your manuscript as you do in your answer to Referee #1?

Regarding the role of the recently increased river inputs of nutrients, it looks to me that the first paragraph on p.13 contains some contradictory sentences. You start by writing "This means that 1) Sargassum distribution is sensitive to these forcings, but 2) these forcings alone cannot control the total Sargassum biomass. " but then you write "For the contribution of the Amazon, this is in agreement with recent conclusions from Johns et al. (2010) and Jouanno et al. (2021)."
First, the reference to Jouanno et al 2021 is missing in the manuscript, I think. Then if I refer to https://www.insu.cnrs.fr/fr/cnrsinfo/proliferation-des-algues-sargasses-le-role-des-fleuves-ecarte , it looks to me that you conclude that the rivers have no impact; and this seems contradictory to me to the sentence in the current manuscript "This means that 1) Sargassum                distribution                is                sensitive                to                these                forcings,                ...".
Then you go on by writing "the quantification of the role of these different forcing ... remains an open question that will deserve further attention. " So I am not sure what your conclusion is? Do the forcings have an impact or not? Does this still need to be investigated? Please clarify in your manuscript.

As noted by referee #1 (his comment #5), your findings regarding the (weak) impact of rivers may have to do with the low value of N and P half saturation and you should comment on this in the manuscript.

Finally, following Referee #2's comment about the impact of the eddy-permitting resolution you used, please add in your manuscript some of the discussion you provide in your answer to Referee #2.

Thank you for considering these comments.

Dear editor

Thanks for your careful reading of our response and for your suggestions. We have taken into account all of your suggestions, and recognize that they should help the reader to better understand our approach and its limitations.

Regarding the causes of the heavy proliferation simulated in March-June we add the following sentence to section 4.1 : "The model tends to reproduce heavy proliferations in March-June which seem not to be observed. Given current knowledge, it is difficult to determine the causes of such a bias. It could be due to a bias in the nutrient content simulated by PISCES-Q at this period. Moreover, error in the Sargassum initial conditions (January) and in the transport parameterization can lead to this production too far north during March-June. An observation

bias cannot be ruled out either since this area is very cloudy and present very contrasted *Sargassum* aggregation properties."

We acknowledge that our statement on the Amazon contribution was not clear. In order to clarify our view, we modified the corresponding section as follows: "This suggest that these forcings alone cannot fuel the total *Sargassum* biomass. Regarding the nutrient brought by the Amazon, this is in agreement with recent conclusions from Johns et al. (2020) and Jouanno et al. (2021), who suggest that the riverine fertilization of the Tropical Atlantic is not at the origin of the phenomenon nor control its year-to-year variability. At this stage, the processes controlling the interannual variability and overall increase of Sargassum remains an open question that will deserve further attention. Application of the numerical tracer method initially proposed by Ménesguen et al. (2006), which tracks nitrogen or phosphorus from any source throughout the biogeochemical network, could help identify the nutrient sources that control the phenomenon without altering the large-scale biogeochemical content."

We add the missing reference to Jouanno et al. (2021) in the reference list : *Jouanno, J., Moquet, J. S., Berline, L., Radenac, M. H., Santini, W., Changeux, T., ... & N'Kaya, G. D. M. (2021). Evolution of the riverine nutrient export to the Tropical Atlantic over the last 15 years: is there a link with Sargassum proliferation?. Environmental Research Letters, 16(3), 034042.*

Regarding comment #5 by reviewer1 and the (weak) impact of rivers that may have to do with the low value of N and P half saturation we add the following sentence in the discussion section : "Moreover, it is worth mentioning that the N/P half saturation constant obtained from the basin scale optimization procedure are low (likely because the biogeochemical model tends to have low surface nutrient concentrations in the northern Tropical Atlantic). This could limit the sensitivity of the model to high nutrient inputs."

In the discussion section we now comment on the possible impact of model resolution: "Transport properties may also be impacted by the numerical choices and model resolution. Our model resolution is intermediate ($\sim$ eddy permitting), so we lack some energy at the mesoscale. Since this mesoscale is particularly important for the dynamics in the Caribbean, Gulf of Mexico, or the North Brazil current area, we would expect more realistic transport properties at higher resolution. But our experience is that ¼° NEMO simulations work well in the region on many aspects of the regional dynamics, such as river plume extent (Hernandez et al. 2016, 2017), large scale currents (Kounta et al. 2018), biogeochemistry (Radenac et al. 2020), salinity large scale distribution (Awo et al. 2018) among other. One reason is that the scales of variability in the tropics are larger than at midlatitudes. This is a posteriori confirmed by the present study since we show that the simulated ocean dynamics are good enough to represent the accumulation of Sargassum in the ITCZ, the advection in the Caribbean through the Antilles, and the episodic shedding of Loop Current eddies in the Gulf of Mexico. We also expect that model resolution is only part of the story regarding the dependence of the transport properties to numerics. Surface transport also depends on the vertical resolution of the model in the mixed-layer, the vertical mixing scheme, the degree of coupling of the ocean circulation with the atmosphere or the waves, the wind product used to force the model, etc… In our model, the windage transport coefficient acts as an empirical factor that compensates lacking the explicit simulation of some of these processes and probably helps us to properly simulate a realistic large scale Sargassum advection. Overall, we definitely need to rely on dedicated Lagrangian studies such as the one performed by Putman et al. (2018), Putman and He (2013),

Berline et al. (2020), Putman et al. (2020) to better constrain our model, and learn about best practices in terms of forcing sargassum transport."

Julien Jouanno on behalf of the authors